# Identification of a Glycosylated Fraction Involved in Mushroom Off-Flavors in Grapes: Influence of Gray Rot, Powdery Mildew and *Crustomyces subabruptus*

**DOI:** 10.3390/molecules27217306

**Published:** 2022-10-27

**Authors:** Léa Delcros, Teddy Godet, Sylvie Collas, Marion Hervé, Bruno Blondin, Aurélie Roland

**Affiliations:** 1MHCS, 51200 Epernay, France; 2Comité Champagne, 51200 Epernay, France; 3UMR SPO, Université de Montpellier INRAE, Institut Agro, 34060 Montpellier, France

**Keywords:** mushroom off-flavor, glycosides, 1-octen-3-one, 1-octen-3-ol, 3-octanol, molds

## Abstract

An organoleptic defect, termed fresh mushroom off-flavor and mainly caused by the C8 compounds 1-octen-3-one, 3-octanol and 1-octen-3-ol, has been identified in wines and spirits since the 2000s. The aim of this work was to identify the presence of glycosidic precursors of these C8 compounds and to evaluate the influence of different molds on the glycosylated fractions of three grape varieties. Must samples contaminated by molds (gray rot, powdery mildew and *Crustomyces subabruptus*) and three levels of attack severity (from healthy to 10–15%) were studied. After a *β*-glycosidase treatment on Meunier and Pinot noir musts contaminated by *Crustomyces subabruptus*, 1-octen-3-one, 1-octen-3-ol and 3-octanol were identified by GC-MS, proving the existence of glycosidic fractions in the musts. A Pinot noir must contaminated by *Crustomyces subabruptus* displayed a 230% increase in the glycosylated fraction responsible for 1-octen-3-one in comparison with an uncontaminated sample. Powdery mildew did not appear to affect the levels of the studied glycosidic fractions in Chardonnay musts. Gray rot on Meunier and Pinot noir musts had opposite effects depending on glycoside type, decreasing the 1-octen-3-one fraction and increasing the 1-octen-3-ol fraction.

## 1. Introduction

In the Champagne wine region, since 2005, an organoleptic defect reminiscent of fresh mushroom has been appearing in some wines. In this region, this defect is mainly found in Pinot noir and Meunier wines. This aroma is thought to be due to molds that accompany the development of gray rot on berries during grape ripening. From a sensory point of view, the defect is not detected in musts but rather appears at the end of alcoholic and malolactic fermentations as reported by different observations by winegrowers [1].

The biogenesis of fresh mushroom off-flavors in food and beverages is complex, and the first publication dealing with this topic dates back to 1982, where 1-octen-3-ol was found responsible for the fresh mushroom/metallic aroma in shellfish [2]. In food items such as banana, soybean oil, fish or cooked meat and mushrooms, compounds with unsaturated aliphatic chains of eight or nine carbon atoms such as 1-octen-3-ol, 1-octen-3-one and 1-nonen-3-one were identified as being responsible for this aroma [3,4,5].

In enology, for the first time in 1980, 3-octanol and 1-octen-3-ol, two molecules with a fresh mushroom odor, were identified in Pinot noir wines [6]. From a sensory point of view, perception thresholds in model wines are quite low, close to 0.04 µg/L for 1-octen-3-one, 40 µg/L for 1-octen-3-ol and 0.03 µg/L in red wine for nonen-3-one [7,8]. In terms of concentration, 1-octen-3-one ranges from 20 to 180 ng/L in altered Champagne base wines (wines before second alcoholic fermentation) [8] and from 0.01 to 0.13 µg/L in fungus-contaminated grape musts of Cabernet Sauvignon and Merlot [7]. For 1-octen-3-ol, the concentration varies between 6 and 21 µg/L in musts obtained from rotten Cabernet Sauvignon, Merlot, Gamay and Sémillon grapes [7]. Today, according to The Goods Scents Company database, more than 70 molecules are known to be responsible for the mushroom aroma, highlighting the molecular complexity of this descriptor. To understand the origin of such a defect, some studies investigated the influence of fungus contamination from the vineyard to the final product. In 1998, the presence of 1-octen-3-ol in musts was associated for the first time with grape sanitary status and more particularly with the presence of *Botrytis cinerea* [9]. Both 1-octen-3-ol and 1-octen-3-one were identified in Cabernet Sauvignon and Sauvignon blanc grapes and grape juice contaminated by various *Ascomycetes* fungi, including *Erysiphe necator*, the fungus responsible for powdery mildew [10]. A possible increase in the quantity of 1-octen-3-ol as a function of the severity of powdery mildew attack on Chardonnay grapes and musts was noted; however, the compound was no longer found in wines from these contaminated grapes [11]. Both 1-octen-3-ol and 1-octen-3-one were also found in Riesling, Gewurztraminer [12], Merlot, Gamay, Semillon and Cabernet Sauvignon musts affected by *Botrytis cinerea*, the gray rot pathogen, and *Penicillium brevicompactum* [7]. In the case of musts affected by *Botrytis cinerea*, the associated Fiano wines contain a concentration of 1-octen-3-ol that increases with the percentage of rotten grapes [13]. Recently, it was demonstrated that *Crustomyces subabruptus* was able to release 1-octen-3-one in both contaminated musts and wines under laboratory conditions [14].

In the fungi kingdom, the biosynthesis of C8 molecules is well-documented. They originate from the transformation of linoleic acid under the action of several enzymes, possibly different lipoxygenases and hydroperoxide lyases [15,16]. From a biological viewpoint, these molecules are used by fungi to regulate their reproduction and as a defense mechanism [17]. In grapes or wine, however, our knowledge of the biosynthetic pathways of these compounds is still quite limited. Nevertheless, the fluctuating characteristics of fresh mushroom off-flavor during the winemaking process suggest that odorless precursors of these molecules might already be present in musts. Glycosidic precursors are a diverse group of odorless compounds first identified in grapes [18]. These molecules are always composed of an osidic moiety connected by a glycosidic bond to a volatile molecule, termed aglycone. When the glycosidic precursor is enzymatically or chemically cleaved, the aglycone moiety is released and its aroma expressed in the wine matrix. Depending on the grape variety, vineyard management and climatic conditions, the proportion of glycosides varies in the grape berry [19]. The main flavors derived from glycosidic precursors are terpenols, C13-norisoprenoids, alcohols, phenol derivatives or C6 compounds. In the literature, 1-octen-3-ol was isolated as glycosides in several species, including *Trifolium subterraneum* [20], *Mentha spicata* [21], *Salvia dichroantha* [22] or more recently in soybean [23]. In addition, several grape varieties, such as Merlot [24] and Falanghina [25] or Fiano grape juice [26] and Chardonnay wines [27], exhibit the presence of 1-octen-3-ol after *β*-glycosidase treatment, suggesting the presence of glycosidic precursors for this specific molecule. Moreover, Chardonnay is a grape variety whose glycosidic fractions have been studied on several occasions [28,29,30,31].

However, to our knowledge, no studies dealing with the correlation between the known molecules implicated in fresh mushroom off-flavors, glycosides levels and type of rot have been conducted and even less so in Pinot noir and Meunier grapes. Consequently, the first objective of this study was to formally identify the presence of glycosidic precursors that can release aglycones with a mushroom off-flavor, with a focus on 1-octen-3-one, 1-octen-3-ol and 3-octanol, using laboratory model conditions. The second aim was to observe the influence of powdery mildew affecting Chardonnay and gray rot contaminating Meunier and Pinot noir on the proportions of the glycosidic fractions of 1-octen-3-one, 1-octen-3-ol and 3-octanol.

## 2. Materials and Methods

### 2.1. Chemicals

Hydrogen di-sodium phosphate (99%) and methanol (LC-MS grade > 99.9%) were purchased from Fisher Scientific (Illkirch, France). Citric acid (99%), dichloromethane (≥99.9%), pentane (≥99%) and sodium sulfate anhydrous (≥99%) were purchased from Sigma Aldrich (St Quentin Fallavier, France) as well as 4-nonanol (≥96.5%), which was used as an internal standard. Sulfites were purchased as a commercial 10% (*w*/*w*) solution from a local distributor. Pure 1-octen-3-one (97%), 1-octen-3-ol (98%), 3-octanol (99%) and 1-nonen-3-ol (98%) standards were purchased from Fisher Scientific (Illkirch, France). Enzymatic kits were purchased from Neogen Megazyme (Scotland, UK) and Rapidase Revelation Aroma enzyme from DSM (Montpellier, France).

### 2.2. Grape sampling and Must Preparation

#### 2.2.1. Grape Sampling

The different grapes were collected in Champagne (France) during the 2019 and 2020 harvests. The different samples originated from 3 plots of Chardonnay (plots A, B C), 2 plots of Meunier (D, E) and 1 plot of Pinot noir (F), as summarized in Table 1.

#### 2.2.2. Must for the Laboratory Contamination Model

Two batches of healthy grapes were hand-picked (one of 160 kg of Meunier and one of 160 kg of Pinot noir) during the 2019 and 2020 harvests, respectively. Grapes were pressed at Comité Champagne in a pneumatic press according to the extraction programs required by the appellation. The resulting juice (85 L) was treated with pectolytic enzymes (E Catal^®^ Clarif, Station Oenotechnique de Champagne, Cormontreuil, France, 1 g/hL) before being divided into a 20 L batch (A) and a 65 L batch (B). Sulfites were added at 3.5 g/hL to batch B, while batch A was kept sulfite-free. After temperature-controlled settling (15 °C, 12 h), the sulfite-free must (A) was contaminated with *Crustomyces subabruptus* according to Meistermann et al.’s protocol [14]. The contaminated must was then diluted to 1/100 with the sulfited must (B) and stored at −20 °C before experiments.

#### 2.2.3. Must from Grapes Affected by Vineyard Molds

Chardonnay, Meunier and Pinot noir grapes were harvested in Champagne vineyards in 2019 and 2020. Batches of 50 kg of visually healthy grapes, 50 kg of grapes with 1–5% attack intensity and 50 kg of grapes with 10–15% attack intensity were collected in the vineyard by considering the visible symptoms of the disease for each plot (A, B, C, D, E, F) of each grape variety like previous works [13,32]. Batches were pressed in a pneumatic press (Europress, Scharfenberger, Germany) according to the extraction programs required by Champagne appellation. The collected musts were sulfited at 6 g/hL for Chardonnay must and at 7 g/hL for Meunier and Pinot noir musts, doses recommended by the Champagne appellation. Musts were treated by pectolytic enzymes (E Catal^®^ Clarif, Station Oenotechnique de Champagne, 1 g/hL) and then settled overnight at 15 °C. The clarified musts were then stored at −20 °C before analysis.

#### 2.2.4. Analysis of Enological Parameters

Table 1 summarizes the different samplings performed on fresh musts and their enological parameters. Sugar content and potential alcoholic strength by volume were estimated by densimetry. Total acidity was measured by potentiometric titration in g H_2_SO_4_/L (OIV-MA-AS313-01). Gluconic acid and glycerol were quantified using enzymatic kits (D-gluconic acid assay kit and glycerol GK assay kit) by UV-visible spectrophotometry (OIV-MA-AS313-28, OIV-MA-AS312-05). All these enological parameters were measured following the protocols established by the International Organization of Vine and Wine (OIV).

### 2.3. Analysis of Glycosidic Precursors

#### 2.3.1. Sample Extraction

The extraction of glycosides was based on the protocol established by Gunata et al. [33] and repeated by Martínez-Gil et al. [31] with some modifications. Each sample extraction was performed in triplicate. Grape must (300 mL) was fined with washed polyvinylpolypyrrolidone (PVPP, 3 g, Sigma Aldrich, Saint-Quentin-Fallavier, France) for 20 min and filtered once on paper (Whatman, Merck, Lyon, France) and a second time on a 5 μm cellulose nitrate filter (Whatman, Merck, Lyon, France). The filtrate was separated in three 100 mL fractions, and glycosides were extracted on SPE cartridges (Lichlorut RP-18 500 mg/3 mL, Merck, Lyon, France). The cartridges were conditioned with methanol (15 mL) and Milli-Q^®^ water (20 mL) before filtrate loading (100 mL). Cartridges were washed with a pentane–dichloromethane mixture (2/1, *v*/*v*, 15 mL), and glycosides were eluted with methanol (8 mL). The extract was concentrated to dryness under vacuum (30 °C, 30 mbars, 3 h, Genevac apparatus) and stored at −20 °C before enzymatic hydrolysis.

#### 2.3.2. Enzymatic Hydrolysis

The dry extract was first resuspended in a citrate–phosphate buffer (hydrogen di-sodium phosphate 0.2 M–citric acid 0.1 M; pH 5; 1 mL). Enzymes (Rapidase Revelation Aroma *α* and *β*-glycosidase) were added (100 µL of a 70 mg/mL solution in the citrate–phosphate buffer) and samples maintained at 35 °C overnight [33]. Released aglycons were extracted five times with a pentane–dichloromethane mixture (2/1 *v*/*v*, 5 × 1 mL) before being dried over Na_2_SO_4_. The 4-nonanol internal standard was spiked (25 µL of a solution at 330 mg/L in absolute ethanol). Aglycon extracts were concentrated (down to 250 µL) using a Dufton column at 40 °C before being analyzed by GC-MS. A few drops of hexane were added to the vial before injection.

#### 2.3.3. Gas chromatography–Mass Spectrometry (GC-MS) Analysis

Analysis of aglycons was performed using an Agilent 6890 gas chromatograph, coupled with a mass spectrometer of the same series. The GC was equipped with a fused silica capillary column (length 30 m × 0.25 mm diameter and 0.5 µm film thickness, DB-Wax, Agilent J&W, Santa Clara, CA, USA). The autosampler was tempered to 6 °C, and the injection volume was 2 µL for the injector set at 245 °C. The oven temperature gradient was set from 40 °C (3 min isothermal) to 245 °C at 3 °C/min, and then isothermal for 10 min. The analytical run lasted 81 min. The transfer line was set at 250 °C and the electronic impact source (EI) temperature kept at 250 °C. The analyses were first performed in Full Scan (35–350 uma) mode and then in Selected Ion Monitoring mode. The compounds were identified by comparison of their mass spectra with those in the spectra library and confirmed by injection of the commercially available pure compounds, for 1-octen-3-one, 1-octen-3-ol, 3-octanol and 1-nonen-3-ol. The Selected Ion Monitoring mode was then used to quantify the compounds as 4-nonanol equivalents, and the ion fragments selected for each analyte are referenced in Table 2. The concentration of each compound was determined using the average of three analytical repetitions.

### 2.4. Analysis of Free C8 Compounds in Grape Must

From 100 mL of grape must (triplicate analysis), the free C8 compounds were extracted three times with pentane–dichloromethane (2/1, *v*/*v*, 3 × 50 mL) before being dried over Na_2_SO_4_. The extracts were distilled (down to 5 mL). Then, 4-nonanol was added (25 µL of a 330 mg/L solution in absolute ethanol), and the extracts were rectified (up to 250 µL) using a Dufton column at 40 °C before being analyzed by GC-MS. Before injection, a few drops of hexane were added to the vial.

### 2.5. Statistical Analysis

JMP Pro 14.2 Statistical Discovery from SAS and Microsoft Excel were used to calculate sample means and standard deviations considering the analytical replicates of the three experiments (n = 3). The effects of powdery mildew and gray rot on the aglycone concentration in must were analyzed using the *t*-test (95%) considering: * significant (*p* < 0.05) and ** highly significant (*p* < 0.01).

## 3. Results and Discussion

The different enological parameters of the musts used are listed in Table 1. Sugar values (as total soluble solids) ranged from 164 to 176 g/L for Chardonnay, from 188 to 202 g/L for Meunier and from 201 to 205 g/L for Pinot noir, regardless of the infection status. The total acidity values varied between 5.8 gH_2_SO_4_/L and 7.8 gH_2_SO_4_/L and pH values between 2.89 and 3.10, regardless of the grape variety and infection level. These different enological parameter values agreed with those observed in the vineyard by Comité Champagne during the 2019 and 2020 harvests. Glycerol and gluconic acid, which are two markers of overall health status, displayed values below detection limits for healthy batches, while the concentrations reached 0.5 g/L for glycerol and 0.52 g/L for gluconic acid for gray-rot-affected ones. These observations are in coherence with data reported by Hausinger et al. [35].

### 3.1. The Laboratory Contamination Model with Crustomyces subabruptus

In order not to depend on climatic conditions in the vineyard, a contamination model based on several years of research has been designed [14]. *Crustomyces subabruptus*, a fungus initially isolated in the Alsace wine region, was also found in Champagne vineyards in 2014 (strain CIVC 02RE14). This fungus, which belongs to the *Cystostereaceae* family, grows inside grape bunches and is able to impart a fresh mushroom off-flavor by producing 1-octen-3-one to wines from a frozen must of healthy grapes [14]. This contamination protocol resulted, in a systematic and reproducible way, in a wine with fresh mushroom aromas and constituted the starting point for this study.

#### 3.1.1. Characterization of free C8 Compounds

First, the aim was to verify the direct release of C8 molecules with fresh mushroom off-flavors in musts resulting from the contamination model [14,36]. For this purpose, a mixture of pure compounds (1-octen-3-one, 3-octanol, 1-octen-3-ol and also 4-nonanol, which was the chosen internal standard) was analyzed in GC-MS (Full Scan detection) to identify the retention times and the characteristic specific ions (*m/z*). All these data are listed in Table 2 as well as the olfactory perception thresholds. 1-octen-3-one has the lowest odor threshold in neutral white wine at 40 ng/L, [8], which is one-thousand-fold lower than the odor threshold of 1-octen-3-ol.

The chromatogram of the healthy Pinot noir must (control), visible in Figure 1A, showed a peak at 23.19 min corresponding to the internal standard, 4-nonanol. Two peaks at 19.65 min and 21.90 min were identified as 3-octanol and 1-octen-3-ol, respectively. However, no peak was detected at the expected retention time for 1-octen-3-one, which is consistent with the fact that the sample was fungus-free. The sample of contaminated Pinot noir must gave the chromatogram presented in Figure 1B, with the three peaks at 15.34 min, 19.65 min and 21.90 min identified as 3-octanol, 1-octen-3-ol and 1-octen-3-one, respectively. The contamination by *Crustomyces subabruptus* resulted in a specific release of 1-octen-3-one in comparison with the control sample as reported by Meistermann [14], with a concentration rising to 19 ± 3 µg/L equivalent of 4-nonanol. Concerning the other compounds, we quantified similar amounts of 3-octanol and 1-octen-3-ol between both samples (control vs. contaminated), close to 0.18 ± 0.03 µg/L equivalent of 4-nonanol. *Crustomyces subabruptus* did not seem to affect the production of these latter two compounds. Thanks to these tests, we validated the laboratory model as a systematic and reproducible tool for 1-octen-3-one production. Consequently, we used this contamination model to investigate the presence of glycosidic precursors of such aglycons in *Crustomyces subabruptus*-infected musts.

#### 3.1.2. Characterization of a Glycosylated Fraction Responsible for Fresh Mushroom Off-Flavors in *Crustomyces subabruptus*

Since free 1-octen-3-one, 3-octanol and 1-octen-3-ol were present in the contaminated Pinot noir model must, the aim of the study was to search for their associated glycosidic fractions. For this purpose, we decided to use an indirect and specific analysis based upon the cleavage of glycosidic bonds by a *β*-glycosidase and the analysis of released aglycons [37]. Indeed, aglycones can be released in two ways, through acid or enzymatic hydrolysis. Acid hydrolysis can cause acid-catalyzed cyclizations, dehydrations and rearrangements [38,39] as well as compositional changes [40]. The enzymatic hydrolysis proposed by Gunata et al. [33] leads to fewer secondary reactions; therefore, this second hydrolysis strategy was selected for this study.

First, to ensure that the volatile molecules released were indeed the result of the specific cleavage of the glycosidic bonds and not a pollution or an artifact, controls without the enzyme addition were analyzed. In practice, the two contaminated musts of the model, Pinot noir and Meunier, were treated without and with the addition of *β*-glycosidase (0 or 100 µL of Rapidase Revelation Aroma of a 70 mg/mL solution in the citrate–phosphate-buffered solution). Figure 2A represents the control sample of Pinot noir must without the *β*-glycosidase addition. No peak was detected at the expected retention times of the molecules of interest, as referenced in Table 2. Thus, the glycoside extraction method, without the enzymatic hydrolysis step, did not produce or release free C8 compounds in the samples. Figure 2B represents a sample of the same Pinot noir must treated with *β*-glycosidase. Several signals were detected confirming the enzymatic cleavage of glycosides releasing volatile compounds. At 15.40 min, a peak with fragments m/z 55; 70; 97 appeared, corresponding to the 1-octen-3-one molecule, by analogy with the NIST library mass spectra and comparison with the previously injected commercial product. At a 19.67 min retention time, 3-octanol was detected (m/z 59; 83; 101) as well as 1-octen-3-ol at 21.95 min. These three molecules therefore originated from glycosides released by enzymatic cleavage in the Pinot noir must from the contamination model. The same analyses were carried out on the Meunier must contaminated by the model, and similar observations were made; none of the C8 compounds of interest could be identified without the addition of *β*-glycosidase, whereas 1-octen-3-one, 3-octanol and 1-octen-3-ol were all detected after the enzyme addition to the glycoside extract (data not shown).

In the literature, 1-octen-3-ol had already been identified as free and glycosidically bound in other grape varieties, such as Merlot [24], Falanghina [25] and in Fiano must [26]. Yet, no data concerning the sanitary status of grapes on the one hand and no reference concerning Pinot noir and Meunier on the other hand are available. Consequently, we investigated the possible presence of glycosidic precursors in healthy Meunier and Pinot noir grapes. Following the same strategy, we clearly identified an aglycon fraction as being responsible for the release of 1-octen-3-one, 3-octanol and 1-octen-3-ol, demonstrating for the first time that glycosidic precursors may be present in healthy Meunier and Pinot noir grapes. In addition, the 3-octanol aglycone had never been reported in grapes and only one reference dealing with Chardonnay wines referred to this aglycon [27]. The formal identification of these three C8 aglycones responsible for fresh mushroom off-flavors in healthy musts implies that the production of these compounds in their glycosidic form might be part of the plant metabolism. Indeed, in soybean, a glycoside of 1-octen-3-ol, 1-octen-3-yl *β*-primeveroside, is immediately hydrolyzed when plant tissues are damaged, releasing 1-octen-3-ol [23]. The physiological functions of 1-octen-3-ol vary depending on the organism, but it has been reported in *Arabidopsis thaliana* that this molecule activates a defensive response of the plant inhibiting the expansion of necrotic lesions caused by *Botrytis Cinerea* [41]. It is therefore possible to assume that the vine has a stock of 1-octen-3-ol glycoside, which, in the presence of a pathogen, hydrolyses, alerting the plant and triggering a defense mechanism against the pathogen.

The proportions of each aglycone in 4-nonanol equivalent per liter of must, presented in Figure 3, are roughly the same in both Meunier and Pinot noir. The 3-octanol aglycon fraction represented more than 50% of the aglycones of these C8 molecules responsible for fresh mushroom aromas, while the 1-octen-3-ol aglycone represented about 30% of these C8 molecules. When *Crustomyces subabruptus* contaminated the Meunier must, the amount of aglycones of the three C8 molecules significantly increased by about 19% (Figure 3). The amount of 1-octen-3-one glycosylated fractions increased from 0.02 to 0.04 ± 0.01 μg/L of 4-nonanol equivalent between healthy and contaminated Meunier must (100% increase). The quantity of 3-octanol glycosylated fractions increased by 3.6% and that of 1-octen-3-ol by about 42%. In the contaminated Pinot noir must, only the amount of 1-octen-3-one aglycone increased, while the other compounds remained at approximately the same concentrations as in the healthy Pinot noir must. The 1-octen-3-one aglycone concentration increased by about 230% in the contaminated Pinot noir must compared to a healthy sample. The glycosylated part of 1-octen-3-one that could originate from usual grape metabolism represented about 30% of the value measured in the sample contaminated by *Crustomyces subabruptus*, which may suggest that this fungus was able to increase the glycosidic potential of C8 molecules.

As an intermediate conclusion, the *Crustomyces subabruptus* contamination model released 1-octen-3-one in its free form in a must, as predicted by Meistermann et al. [14], but the release of 1-octen-3-one in a glycoside-bound form in Meunier and Pinot noir musts was observed here for the first time.

### 3.2. Influence of Diseases on Glycoside Fractions on Different Musts

The laboratory model established with *Crustomyces subabruptus* allowed the detection of 1-octen-3-one, 3-octanol and 1-octen-3-ol glycosides in grape musts. In Champagne vineyards, the most common grape diseases are powdery mildew and gray rot. The next part of the study tried to evaluate the influence of these diseases on the glycosylated fractions of C8 molecules. For this purpose, batches of Chardonnay, Meunier and Pinot Noir grapes affected by gray rot or powdery mildew were visually constituted at different attack severity levels, from healthy to 1–5% and 10–15% of attack severity. Due to vineyard conditions, only Chardonnay grapes were affected by powdery mildew in the 2019 and 2020 vintages, explaining why our dataset only focused on this grape variety for this particular disease.

#### 3.2.1. Influence of Powdery Mildew on Chardonnay Musts

Aglycones of 1-octen-3-one, 3-octanol and 1-octen-3-ol were found in each must from each Chardonnay plot whatever the level of infection (Table 3). This is in agreement with the work of Hampel et al. [27], who had already identified the presence of 3-octanol and 1-octen-3-ol glycoside in Chardonnay berry skin. Glycosylation is a natural biological process in vine that correlates with these observations. The three plots studied seemed to have roughly the same profile of 1-octen-3-one aglycone amounts, i.e., between 0.02 and 0.04 ± 0.02 µg/L 4-nonanol equivalent. With regard to plots A and C, which were two different plots from the 2019 and 2020 harvests, powdery mildew had no influence on the production of glycosylated fractions of 1-octen-3-one whatever the level of visual attack. Only plot B, harvested in 2019, showed a significant increase in the amount of 1-octen-3-one glycoside for the 10–15%-affected sample. Below this attack severity threshold, the increase was not significant. In the literature, Darriet et al. [10] identified free 1-octen-3-one in Sauvignon blanc grapes affected by powdery mildew. By comparing our findings with these, we can assume that 1-octen-3-one could be present both as a glycosylated precursor and also in its free form in powdery-mildew-affected white grape juice, but it is still difficult at the moment to correlate the disease level with the amount of this C8 compound. The glycosylated fractions of 3-octanol and 1-octen-3-ol did not change with powdery mildew severity in the three plots observed, suggesting that there was no influence of this specific alteration on these specific glycosides. However, Rusjan et al. [11] showed an impact of powdery mildew on the free 1-octen-3-ol molecules in Chardonnay musts, in which the 1-octen-3-ol concentration increased significantly as a function of attack severity. However, the four degrees of powdery mildew attack intensity reported by Rusjan et al. [11] were 0%, 10%, 50% and 100%, two of which were considerably higher than those used in our study; this could explain why no powdery mildew influence could be detected under our conditions.

#### 3.2.2. Influence of Gray Rot on Meunier and Pinot Noir Musts

One of the diseases frequently encountered in Champagne vineyards is gray rot caused by a fungus termed *Botrytis cinerea*. This disease primarily affects the two Champagne black grape varieties, Pinot noir and Meunier, but can also be found on Chardonnay. Here, the study focused on Meunier musts (plots D and E) and Pinot noir musts (plot F) affected at different degrees with grey rot. First, glycosylated fractions of the three C8 molecules investigated were detected in all the samples (Table 3).

Concerning 1-octen-3-one glycosylated fractions, concentrations decreased significantly as soon as the level of contamination reached at least 1% for all the samples. By analogy, Pinar et al. [12] identified free 1-octen-3-one in Riesling and Gewurztraminer musts affected by *Botrytis cinerea*. The amount of free 1-octen-3-one seemed to increase in botrytized musts compared to healthy musts. *Botrytis cinerea* possesses *β*-glycosidases that are expressed both intra- and extracellularly [42]. According to one hypothesis, *Botrytis cinerea* might be involved in the cleavage of 1-octen-3-one glycoside precursors through its own *β*-glycosidases and could directly induce this degradation during the wine-making process.

Regarding 1-octen-3-ol aglycones, the concentration of glycosylated fractions increased with the gray rot attack severity (two plots over three) and seemed specific to Meunier only. No effect was observed in Pinot noir must, but our dataset is too limited to make conclusions about this grape variety. Moreover, there are no studies dealing with glycosidic precursors in Meunier grapes, which prevents us from comparing and correlating these observations with other research works. Following the hypothesis in Section 3.1.2 and the observations of this study, grape musts produced from grapes affected by gray rot have a higher amount of 1-octen-3-ol glycoside. The hypothesis made in Section 3.1.2 envisages that these glycosides allow the induction of a defense mechanism of the vine against a pathogen, such as *Botrytis cinerea*. It is conceivable that when the first grape berries are affected by *Botrytis cinerea*, the glycosides release 1-octen-3-ol, alerting the vine to the presence of the pathogen. In order to safeguard the uninfected berries and counteract the pathogen by preventing it from spreading across the berries; by analogy with *Arabidopsis thaliana* [41], the grapevine would then produce more and more 1-octen-3-ol glycosides and store them in the still-intact berries. During pressing, these still-healthy berries are pressed and the glycosides are extracted into the must, which explains the higher levels found in the spoiled musts. These extracted glycosides in higher quantities could be cleaved during alcoholic fermentation and thus release more 1-octen-3-ol in the wines. One study showed an increase in the concentration of 1-octen-3-ol in Garganega wines obtained from *Botrytis cinerea* rotten grapes compared to wines obtained only from healthy grapes [13]. However, this remains a hypothesis in this study.

Finally, our experimental conditions did not allow us to make conclusions about the effect of *Botrytis cinerea* on the biosynthesis of glycosylated fractions of 3-octanol. Indeed, just one plot out of three was impacted by the effect of gray rot and showed a significant increase in 3-octanol aglycones (plot D). Indeed, glycoside accumulation depends on the grape variety as well as on the vine’s behavior and climatic conditions. Thus, it is only logical to observe differences between two grape varieties affected by the same disease and between two plots of the same grape variety.

In conclusion, the glycosidic precursors of 1-octen-3-one, 1-octen-3-ol and 3-octanol behave differently in the presence of *Botrytis cinerea*. Indeed, *Botrytis cinerea* could cleave the 1-octen-3-one precursors, thus decreasing their quantity in the musts. For the 1-octen-3-ol glycoside, there are two opposing hypotheses. Firstly, *Botrytis cinerea* itself produces this glycoside, increasing its quantity in contaminated musts, but no bibliographical data currently support this hypothesis. The second hypothesis is that the plant itself produces this glycoside in greater quantities in order to protect itself from *Botrytis cinerea* by activating a defense process. Indeed, when glycosides are cleaved and release 1-octen-3-ol, the expansion of *Botrytis cinerea* could potentially be stopped. For the glycosidic precursors of 3-octanol, it is difficult to propose a hypothesis regarding its variation in musts as there are few data in the literature and the observations of the study seem to be plot-dependent.

## 4. Conclusions

This work allowed the identification in Meunier, Pinot noir and Chardonnay musts of the unambiguous presence of 3-octanol as well as 1-octen-3-ol glycosides and for the first time of 1-octen-3-one glycosides. *Crustomyces subabruptus*, used under laboratory contamination conditions, significantly increased the concentrations of 1-octen-3-one glycosides in contaminated Meunier and Pinot noir musts. Powdery mildew (with attack intensities of 0, 1–5 and 10–15%) did not seem to affect the levels of 1-octen-3-one, 3-octanol and 1-octen-3-ol glycosides in Chardonnay musts. Gray rot had opposite effects on Meunier and Pinot noir depending on the type of glycosides; indeed,1-octen-3-one glycosides decreased, while 1-octen-3-ol glycosides increased in musts affected by gray rot. In conclusion, the identification of a glycosylated fraction able to release fresh mushroom aroma compounds was the first step to better understand the source of this specific taint and may help winemakers to avoid off-flavors in the future.

## Figures and Tables

**Figure 1 molecules-27-07306-f001:**
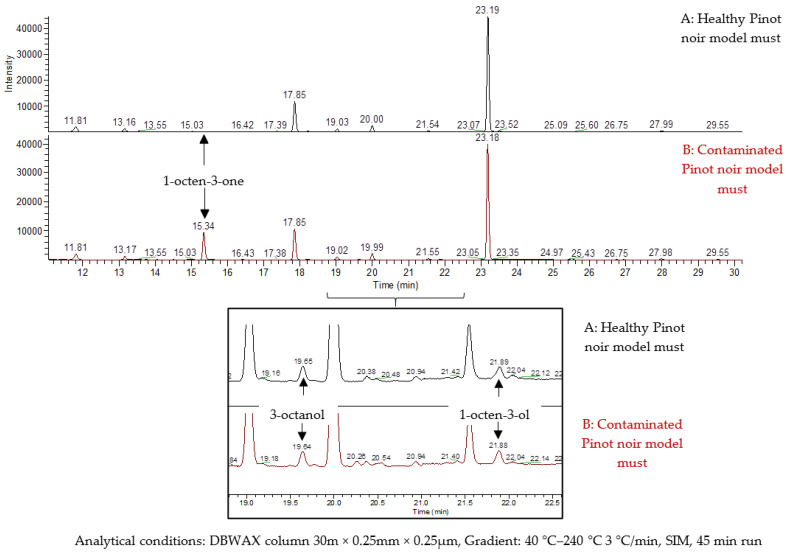
Chromatogram of free C8 compounds of the (A) healthy and (B) contaminated Pinot noir model must.

**Figure 2 molecules-27-07306-f002:**
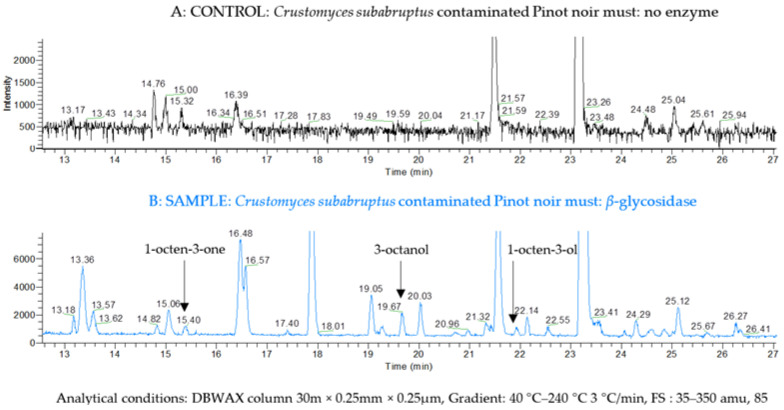
Chromatogram of compounds released from the must of the contaminated Pinot noir model with and without enzyme addition. (**A**) CONTROL: Crustomyces subabruptus contaminated Pinot noir must: no enzyme. (**B**) SAMPLE: Crustomyces subabruptus contaminated Pinot noir must: β-glycosida.

**Figure 3 molecules-27-07306-f003:**
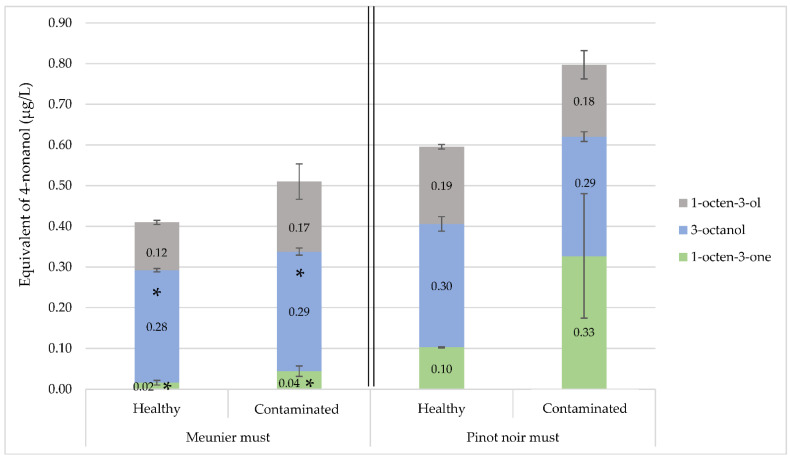
Concentrations of 1-octen-3-one, 3-octanol and 1-octen-3-ol in healthy and contaminated laboratory models (1% *Crustomyces subabruptus* contamination) Meunier and Pinot noir must. Meunier must: *p*-value 1-octen-3-ol = 0.0952, *p*-value 3-octanol = 0.0375, *p*-value 1-octen-3-one = 0.0238; Pinot noir must: *p*-value 1-octen-3-one = 0.0642; *p*-values obtained by *t*-test (95%) between healthy and 1%-contaminated must; * significant (*p* < 0.05); analyses done in triplicate, error bars represent analytical standard deviation (n = 3).

**Table 1 molecules-27-07306-t001:** Sample description and enological parameters.

Sample Description (Vintage)	**Plot A: Chardonnay with** **powdery mildew (2019)**			**Plot B: Chardonnay with** **powdery mildew (2019)**			**Plot C: Chardonnay with** **powdery mildew (2020)**		
Level of mold contamination	Healthy	1–5%	10–15%	Healthy	1–5%	10–15%	Healthy	1–5%	10–15%
Sugar (g/L)	173	167	164	176	172	169	176	175	176
Alcohol (% vol.)	10.3	9.9	9.7	10.5	10.2	10.0	10.5	10.4	10.5
Tot. acidity (g H_2_SO_4_/L)	6.3	7.7	8.1	6.3	7.8	8.0	6.1	6.1	5.8
pH	3.02	2.97	2.94	2.93	2.92	2.94	3.08	3.08	3.07
Glycerol (g/L)	<LOD	<LOD	<LOD	<LOD	<LOD	<LOD	<LOD	<LOD	<LOD
Gluconic acid (g/L)	<LOD	<LOD	<LOD	<LOD	<LOD	<LOD	<LOD	<LOD	<LOD
Sample description (vintage)	**Plot D: Meunier with gray rot (2019)**			**Plot E: Meunier with gray rot (2019)**			**Plot F: Pinot noir with gray rot (2019)**		
Level of mold contamination	Healthy	1–5%	10–15%	Healthy	1–5%	10–15%	Healthy	1–5%	10–15%
Sugar (g/L)	188	194	198	198	199	202	205	202	201
Alcohol (% vol.)	11.2	11.5	11.7	11.8	11.8	12.0	12.2	12.0	11.9
Tot. acidity (g H_2_SO_4_/l)	7.2	7.2	6.9	6.1	5.9	6.1	7.6	7.6	7.8
pH	2.94	2.96	2.98	3.09	3.10	3.11	3.08	3.08	3.07
Glycerol (g/L)	<LOD	<LOD	<LOQ	<LOD	<LOQ	0.4	<LOD	<LOQ	0.5
Gluconic acid (g/L)	<LOD	0.05	0.14	<LOD	0.12	0.52	<LOQ	0.08	0.15
Sample description (vintage)	**Must of model of contamination:** **Meunier (2019)**				**Must of model of contamination:** **Pinot noir (2020)**				
Level of mold contamination	Healthy		1% *Crustomyces subabruptus*		Healthy		1% *Crustomyces subabruptus*		
Sugar (g/L)	199		200		173		173		
Alcohol (% vol.)	11.8		11.9		10.3		10.3		
Tot. acidity (g H_2_SO_4_/L)	7.1		7.5		9.5		7.2		
pH	3.10		3.04		2.97		2.91		
Glycerol (g/L)	<LOQ		0.4		<LOD		n.a.		
Gluconic acid (g/L)	0.04		0.10		<LOD		n.a.		

LOD = limit of detection; LOQ = limit of quantification, n.a. = not available. Enological analyses performed once.

**Table 2 molecules-27-07306-t002:** Retention time, quantification and qualification ions of pure analytical C8 compounds analyzed by GC-MS.

Compounds	Retention Time (min)	Quantification Ion m/z	Qualifier Ion m/z	Perception Threshold (ng/L)
1-octen-3-one	15.34	55	70, 97	40 ^a^ [8]
3-octanol	19.69	59	83, 101	250 ^b^ [34]
1-octen-3-ol	21.90	57	72, 85	40 000 ^a^ [7]

^a^ Perception thresholds determined in neutral white wine. ^b^ Perception thresholds determined in air.

**Table 3 molecules-27-07306-t003:** Concentration in µg equivalent of 4-nonanol of 1-octen-3-one, 3-octanol and 1-octen-3-ol aglycones in musts affected by powdery mildew and gray rot.

		Concentration of Different Aglycons (µg/L Equivalent 4-nonanol) ^#^											
Compounds		1-octen-3-one				3-octanol				1-octen-3-ol			
		Healthy	1–5%	10–15%	SD|PI	Healthy	1–5%	10–15%	SD|PI	Healthy	1–5%	10–15%	SD|PI
Powdery mildew	Plot A Chardonnay (2019)	0.04 ± 0.02	0.04 ± 0.01	0.04 ± 0.01	/	0.21 ± 0.02	0.22 ± 0.01	0.25 ± 0.01	/	0.41 ± 0.12	0.21 ± 0.06	0.36 ± 0.07	/
	Plot B Chardonnay (2019)	**0.02 ± 0.02**	0.04 ± 0.01	**0.07 ± 0.01**	* | ↑	0.27 ± 0.02	0.29 ± 0.01	0.29 ± 0.01	/	0.15 ± 0.03	0.10 ± 0.04	0.14 ± 0.01	/
	Plot C Chardonnay (2020)	0.03 ± 0.03	0.03 ± 0.02	0.04 ± 0.03	/	0.29 ± 0.01	0.30 ± 0.01	0.31 ± 0.01	/	0.18 ± 0.06	0.19 ± 0.02	0.21 ± 0.11	/
Gray rot	Plot D Meunier (2019)	**0.04 ± 0.01**	0.01 ± 0.01	**0.01 ± 0.01**	** | ↓	**0.28 ± 0.00**	0.29 ± 0.01	**0.32 ± 0.01**	** | ↑	**0.18 ± 0.04**	0.20 ± 0.02	**0.25 ± 0.01**	** | ↑
	Plot E Meunier (2019)	**0.03 ± 0.01**	0.02 ± 0.00	**0.02 ± 0.00**	/	0.30 ± 0.01	0.31 ± 0.01	0.30 ± 0.00	/	**0.12 ± 0.01**	0.15 ± 0.01	**0.16 ± 0.01**	** | ↑
	Plot F Pinot Noir (2019)	**0.07 ± 0.01**		**0.02 ± 0.00**	** | ↓	0.30 ± 0.00		0.30 ± 0.00	/	0.24 ± 0.01		0.22 ± 0.02	/

* ^#^ Means of triplicate analysis and ± analytical standard deviation (n = 3). SD = statistically significant differences. Healthy must and must with 10–15% attack intensity (*t*-test at 95%). / = not significant, * significant (*p* < 0.05), ** highly significant (*p* < 0.01). PI = possible impacts of the infection on the aglycon fractions (↑: increase, ↓: decrease). In bold characters for statistically significant effect.

## Data Availability

Not applicable.

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
