# Peer review of "Identification of a Glycosylated Fraction Involved in Mushroom Off-Flavors in Grapes: Influence of Gray Rot, Powdery Mildew and Crustomyces subabruptus"

_molecules, 2022, doi:10.3390/molecules27217306_

Round 1

Reviewer 1 Report

attached

Reviewer 2 Report

This paper is designed to investigate the influence of grey rot, powdery mildew and Crustomyces subabruptus on mushroom off-flavors in grapes, and confirm the existence of a glycosylated fraction.

The study is meaningful and the presentation is well organized, but the manuscript should be improved before publication.

(1) The abstract is not clear. There is no result of grey rot and powdery mildew (different levels).

(2) Statistical significances should be checked, there should be no significance when p<0.1. 

Line 26-32. 1-2 references should be needed at suitable place.

Line 353. How much the sulfite-free must (A) was contaminated?

Line 361. If only one pneumatic press was used, was the potential contamination among different samples considered?

Line 363. Why different amounts of sulfurs were treated?

Table 1. The result data should be mean ± standard deviation.

Line 117-125. This part should be moved to section 3.2.2.

Reviewer 3 Report

Dear Authors,

The article brings to attention a research on a particular production niche in winemaking which contributes through the clarifications presented to a better quality management of these products.

From begin, the authors should respect the indications of the format of the "Molecules" in the presentation of the sections, namely to rearrange the presentation with the inclusion of the materials and methods chapters used before the results chapter in order to have a logical picture in the study presentation.

In subchapter 2.1.1 the characterization of free C8 compounds, the authors should supplement the data from the specialized literature with recent research on this topic, there is an acute lack of examples of more recent research and reporting, 2011 is not very recent, even if still the study from 2021 is presented. That is why I suggest to have more clarity to expand with more examples.

In the following, there is the same need for updating the studies in the specialized literature. Thus in subchapter 2.1.2. data from 1989 and 1985 are cited, but for an increased credibility of the study, other examples of more recent studies can be presented that must be introduced to support the research in this article. I support these requirements to update the data from the specialized literature in that almost half of the 36 bibliographic titles are from the period before the year 2000, with some examples from 2003-2004.

For a better picture, I suggest the authors to fill in the data for the contaminated product in figure 3, 1% is not enough, as specified for the first chromatogram representing the "healthy" fraction, to add it to the other "contaminated" chromatogram.

Considering the authors' choice for the "Food chemistry" section, to support this affiliation, the authors should include in chapter 3.3 (analysis of glycosidic precursors) a brief presentation of the chemical structure of the compounds highlighted in this study, as well as the hydrolysis reactions the enzymes they support.

Best regards

Reviewer 4 Report

Identification of a glycosylated fraction involved in mushroom off-flavors in grapes: Influence of grey rot, powdery mildew 3 and Crustomyces subabruptus

L30-31: The defect is not detected in musts but rather appears at the end of alcoholic and malolactic fermentations as reported by different observations by vinegrowers (Ref is missing here).

L 80: In the literature,

L101: 164 g/L to 176 g/L : Put only unit on the last number. 164 to 176 g/L. Correct this mistake in the entire manuscript

Figure 3: 1%. It was better to repeat the experiment as the y-error bar is too big.

L 315: Replace “So” by Thus

L348: harvests, respectively

L361: [36].Batches : Put space before batches

L390: Enzymatic hydrolysis

This procedure has also to be done in triplicate. The sentence related to this is missing in 3.3.2.

L421: 40°C : Put space after 40

L437: 0%, 1-5% and 10-15%) : Put only % on last number, i.e. delete % on 0 and 5, and remain with the one on 15.

The results are good but not well/fully discussed. Some results are not discussed, and others discussed by comparison with only one report/study. The accent has to be put on this.

Round 2

Reviewer 1 Report

see attached

Author Response

Dear reviewer, 

We modified the manuscript regarding fungal contamination in line 133-134 as follows "Batches of 50 kg of visually healthy grapes, 50 kg of grapes with 1-5% attack intensity and 50 kg of grapes with 10-15% attack intensity were constituted in the vineyard by considering the visible symptoms of the disease for each plot (A, B, C, D, E, F) of each grape variety like previous works [13,32]. "

Author Response

Dear reviewer,

Thank you for your attention to our changes. 

Reviewer 3 Report

Dear Authors,

I have taken note of the changes you have made to my suggestion and thank you for this, it gives a superior clarity to your study.

To my last suggestion, you replied that it is not the purpose of the study to identify the hydrolyzed fractions from a structural point of view, and I agree, but it was a pertinent observation considering your chosen affiliation to "food chemistry".

Best regards

Author Response

Dear reviewer,

Thank you for your attention to our manuscript, we take note of this remark for future work. 

Best regards

Reviewer 4 Report

The manuscript was improved as suggested

It can be accepted. The authors have to re-read the MS to correct errors of spaces, to check singular/plural of word, etc. 

Author Response

Dear reviewer,

Thank you for your attention to our manuscript. We added the missing spaces in line 201. 

Best regards